# Quantifying Changes in Dexterity as a Result of Piano Training in People with Parkinson’s Disease

**DOI:** 10.3390/s24113318

**Published:** 2024-05-22

**Authors:** Hila Tamir-Ostrover, Sharon Hassin-Baer, Tsvia Fay-Karmon, Jason Friedman

**Affiliations:** 1Department of Physical Therapy, Faculty of Medical & Health Sciences, School of Health Professions, Tel Aviv University, Tel Aviv 6997801, Israel; hilatamir@gmail.com; 2Movement Disorders Institute and Department of Neurology, Chaim Sheba Medical Center, Tel Hashomer, Ramat-Gan 5262000, Israel; shassin@tauex.tau.ac.il (S.H.-B.); tsviya.faykarmon@sheba.health.gov.il (T.F.-K.); 3Faculty of Medical & Health Sciences, Tel Aviv University, Tel Aviv 6997801, Israel; 4Sagol School of Neuroscience, Tel Aviv University, Tel Aviv 6997801, Israel

**Keywords:** piano, Parkinson’s disease, uncontrolled manifold, dexterity, training, music, sonification, force sensors

## Abstract

People with Parkinson’s disease often show deficits in dexterity, which, in turn, can lead to limitations in performing activities of daily life. Previous studies have suggested that training in playing the piano may improve or prevent a decline in dexterity in this population. In this pilot study, we tested three participants on a six-week, custom, piano-based training protocol, and quantified dexterity before and after the intervention using a sensor-enabled version of the nine-hole peg test, the box and block test, a test of finger synergies using unidimensional force sensors, and the Quantitative Digitography test using a digital piano, as well as selected relevant items from the motor parts of the MDS-Unified Parkinson’s Disease Rating Scale (MDS-UPDRS) and the Parkinson’s Disease Questionnaire (PDQ-39) quality of life questionnaire. The participants showed improved dexterity following the training program in several of the measures used. This pilot study proposes measures that can track changes in dexterity as a result of practice in people with Parkinson’s disease and describes a potential protocol that needs to be tested in a larger cohort.

## 1. Introduction

Evidence suggests that music-based interventions, especially those involving movement, can be beneficial in maintaining motor and cognitive performance in people with Parkinson’s disease (PwP) [1]. Auditory stimuli have been found to be successful vehicles of motor entrainment compared to visual or tactile rhythmic stimuli [2]. Rhythmic Auditory Stimulation (RAS) has, consequently, been studied extensively and has repeatedly shown its effectiveness in improving parameters of gait such as velocity, stride length, and arm swing, as well as in shortening freezing episodes [3] and—in the case of extended RAS training—in reducing falls in PwP [4].

These findings are usually explained as resulting from the fact that the cerebellar–thalamocortical circuits in the brain that support detection and synchronization to regular perceptual events are relatively preserved in PwP, whereas the basal ganglia–thalamocortical network that supports the synchronization of actions to one’s own internal beat is impaired [5]. RAS externalizes the timing for action, thus providing the possibility for matching the movement with a perceptual cue (i.e., beat). However, it is interesting to note that using musical stimuli for RAS has led to improved effects compared with isochronous metronome rhythm both in gait [6,7] and in finger tapping [8], suggesting that this explanation provides a limited account of how auditory events (in general) and music (in particular) can inform and support motor action in PwP.

Instrument playing has also been found to be effective in improving various motor abilities in PwP. Active group music therapy, including instrument playing and singing, significantly improved motor abilities and emotional status compared to traditional physical therapy [9]. In another experiment, six weeks of active music therapy with synchronized handclapping and foot-stomping led to improved overall mobility in PwP [10].

When hypothesizing the beneficial factors that render music an effective rehabilitative tool, two elements stand out. First, the rich multisensory feedback provided by playing music may serve as an enhanced biofeedback mechanism, facilitating rapid error detection and correction. Second, the engaging nature of music playing likely contributes to the success of music-based interventions. Parallel to these, a line of innovative interventions incorporate other techniques, such as the use of VR-based exergames to improve motor capabilities in Parkinson’s disease patients, which have been shown to enhance balance and gait [11] and improve dexterity [12].

Another form of movement entrainment to music is dancing. A substantial number of studies have examined the benefits of dance coupled with music on maintaining motor abilities in PwP. Across studies, a consistent and clinically significant beneficial effect on motor symptoms has been found, especially for tango and tango-related dance [13,14] as well as PD-specific dance forms (e.g., [15]).

Some motor-related benefits of music may also occur in passive listening or imagining music. Thus, for example, increased accuracy of arm and finger movements in PwP has been found following listening to music [16]. In a similar manner, training in mental singing while walking, without actually singing, improved gait in PwP [17].

Lastly, the efficacy of training with auditory feedback using music sonification, i.e., the transformation of real-time movement parameters to sound, has been shown in several studies involving PwP. Thus, sonification helped improve handwriting in PwP [18], and training consisting of action observation plus sonification led to a significant and sustained reduction in gait severity and duration in PwP [19].

The piano’s unique profile makes it particularly promising for musical intervention: it can be played with one hand or both, it is “beginner-friendly”, and it can be played with one finger and its intonation is fixed, yet it provides a wide range of graded finger and wrist manipulations from basic to virtuosic. Moreover, its precise intonation, polyphonic profile (its ability to produce simultaneous sounds), and its immediate sound onset all provide rich auditory feedback to the player’s spatial and temporal performance, creating, in effect, a natural sonification environment superior to most other acoustic musical instruments.

This open-label pilot experiment aims to describe the methods that can be used to quantify changes in dexterity in PwP and to provide pilot data on the efficacy of a novel six-week, piano-based training program in terms of manual dexterity, finger coordination and independence, and functional use of the hand in PwP. In addition, we will assess quality of life. This pilot study will test the feasibility of this approach and will be used to guide the construction of a larger randomized control trial on the use of piano playing for improving dexterity in PwP.

## 2. Materials and Methods

### 2.1. Participants

Details about the participants are provided in Table 1. The participants were recruited from the Movement Disorders Institute at the Sheba Medical Center. Ethical approval was received from the Sheba Medical Center Institutional Review Board, approval number 8407-21-SMC, and the participants were provided written informed consent before taking part in the experiment.

### 2.2. Inclusion and Exclusion Criteria

The inclusion criteria were (1) idiopathic PD patients aged 40–75 years; (2) Hoehn and Yahr stages I to III; and (3) self-reports of some dexterity difficulties.

The exclusion criteria were (1) participation in an ongoing clinical study or clinical study within 30 days prior to this study; (2) atypical parkinsonian syndrome or secondary parkinsonism (e.g., due to drugs, metabolic neurogenetic disorders, encephalitis, cerebrovascular disease, or other degenerative disease), or other neurologic conditions influencing upper limb movement; (3) significant psychiatric symptoms or history; (4) significant or recent experience in piano playing (participants who are proficient piano players, or had studied piano for more than 3 years in the past, or for shorter periods in the last 5 years); (5) other significant fine motor skills (e.g., high proficiency in playing another musical instrument, video games, etc.); (6) Mini Mental Status Examination (MMSE) score below 25; (7) significant sensory deficits, e.g., hearing or sight impairment; (8) unstable medical disorder; and (9) significant postural or action tremor, moderate dyskinesia.

### 2.3. Equipment

Participants were given a 61-key electronic MIDI piano keyboard for the duration of the experiment (Casio CTS100). The keyboard was connected to an Arduino device, which recorded every key press and release and saved it to an SD card with a time stamp for later analysis (the schematics and software for building the device are available online [20]).

### 2.4. Experimental Protocol

#### 2.4.1. Training Protocol

The participants took part in an individually tailored 6-week piano training program, combining structured and supervised weekly training sessions (totaling 6 h) with independent at-home practice. The training sessions were timed to coincide with the patient’s “on” state, i.e., at the peak of their medication’s effectiveness. Throughout the training, medications remained unchanged. Independent practice sessions occurred at prearranged times to ensure adherence and control duration.

Our unique piano training method combined focused finger exercises on the keyboard with a level-appropriate piano repertory. This is in line with the principles of Music-Supported Training (MST)—see, for example, Schneider et al. [21] and Zhang et al. [22].

The functional use of the hand and fingers was thoroughly evaluated at the beginning (week 1) and at the end of the intervention (week 6), using sensor-based and conventional tests. All supervised training sessions were in-person. An assessment of participants’ progress with the assigned piano repertory was conducted on a weekly basis.

#### 2.4.2. Pre- and Post-Tests

The in-clinic tests on weeks 1 and 6 included the following tests:Mini mental test [23] (performed only in the pre-test)Selected items from the MDS-UPDRS:
◦2.4—Eating tasks;◦2.5—Dressing;◦2.6—Hygiene;◦2.7—Handwriting;◦3.3—Rigidity;◦3.4—Finger Tapping.Box and Block Test [24,25] as a measure of gross manual dexterity.Nine-Hole Peg test [26,27] as a measure of finger dexterity. We used a custom-made, instrumented version described in [28].Finger Synergies and Hand Performance test—a finger-pressing ramp task using four fingers (without the thumb), using unidimensional piezoelectric force sensors (model 208C01; PCB Piezotronics Inc., Depew, NY, USA), described in more detail later.PDQ-39 Questionnaire [29]—assesses how often PwP experience difficulties across 8 dimensions of experiences including ADL, cognition, and emotional well-being. We examined the ADL component and the overall score.Quantitative Digitography Test (QDG)—is a measure of the dynamics of finger movements collected using a piano keyboard [30]. We used an adjusted version of the original experimental protocol using our experiment’s MIDI keyboard: Participants were instructed to press two adjacent keys alternately with two fingers (index and middle fingers) for 60 s. They were asked to perform the movement as fast as possible while keeping the alternation of key presses as regular as possible.

#### 2.4.3. Behavioral Tests

To measure finger dexterity, we used the box and block test [24,31] and quantified the number of blocks that the participants moved over the barrier in one minute. In addition, the participants performed the nine-hole peg test [26,27,32], using an instrumented version of the test, which has the same dimensions as the test described in the original paper [32]. The instrumented test records when the pegs are inserted and removed, and when the hand is in the receptacle, allowing the total duration to be subdivided into individual components. For both of these tests, we measured performance in both the left and right hands.

#### 2.4.4. Uncontrolled Manifold Analysis

When performing a redundant task (i.e., with more degrees of freedom than necessary), there are different ways of coordinating the effectors involved in the task. The uncontrolled manifold approach (UCM) [33,34] is a way of dividing the variance observed in the task into good variance (i.e., variance that does not affect task performance), and bad variance (i.e., variance that does affect task performance). We used a finger-pressing ramp task using four fingers (without the thumb), using unidimensional piezoelectric force sensors (model 208C01; PCB Piezotronics Inc.). First, participants were asked to press as hard as they could, to calculate the maximum voluntary contraction (MVC). This was repeated three times, with a 30 s break between each attempt. Then, the participants were shown a ramp on the screen and were instructed to match the sum of the force produced by the four fingers with the template shown on the screen. The cursor moved from the left side to the right side of the ramp over 8 s, and the participant controlled the height of the cursor via the sum of the force of the four fingers, with the bottom and the top of the ramp corresponding to 2.5% and 22.5% of MVC, respectively.

We used a single-trial version of the UCM analysis [35,36,37]. In each trial, we decomposed the variance into “good” variance, which does not affect the outcome measure (the total force, indicated by the height of the cursor on the screen), and “bad” variance, which does. The total force *F_TOT_* is the sum of the forces produced using the four fingers:$$F_{TOT}=\sum_{i=1}^{4}f_{i}$$
where *f_i_* is the force produced using the *i*th finger. The Jacobian, which transforms from changes in finger forces *df* to changes in the total force *dF_TOT,_* is given by
$${dF}_{TOT}=\left[1\;1\;1\;1\right]df.$$

Then, the change in finger force *df*, which does not lead to a change in total force *dF_TOT_*, is given by the null space of the matrix, which are the solutions to the equation
$$0=\left[1\;1\;1\;1\right]e_{i}.$$

The solutions are given by [−1/2 5/6 −1/6 −1/6]^T^, [−1/2 −1/6 5/6 −1/6]^T^, and [−1/2 −1/6 −1/6 5/6]^T^. We projected the change in forces onto these vectors and summed them to find the amount of force that does not affect the outcome measure, given by *f_||._*
$$f_{\left|\right|}=\sum_{i=1}^{3}\left(e_{i}^{T}\cdot df\right)e_{i}$$

The rest of the change in forces, by definition, must affect the outcome measure, $f_{⊥}$*_._*
$$f_{⊥}=df-f_{\left|\right|}$$

We can now calculate the amount of good variance, *v_good_*, by squaring *f_||_* and normalizing its dimension:$$v_{good}=\sum_{i=1}^{N_{samples}}\frac{{\left|f_{\left|\right|}\right|}^{2}}{3N_{samples}}.$$

Bad variance *v_bad_* is defined similarly:$$v_{bad}=\sum_{i=1}^{N_{samples}}\frac{{\left|f_{⊥}\right|}^{2}}{N_{samples}}.$$

Then, the synergy index Δ*v* is given by the difference between the good and bad variance, normalized by the dimension of the space it is calculated in:(1)$$\Delta v=\frac{v_{good}-v_{bad}}{(3v_{good}+v_{bad})/4}.$$

#### 2.4.5. Quantitative Digitography (QDG)

The Quantitative Digitography (QDG) test offers a quantified, sensitive, measurement of finger dexterity and motor function. With its ability to identify subtle motor impairment, QDG has proven particularly valuable in the study of movement disorders, especially Parkinson’s Disease (PD) [30]. The test involves fast alternate finger tapping (using the index and middle fingers) on a MIDI keyboard for 60 s. The recorded data are then analyzed for tapping speed, rhythm, finger independence, the force exerted by each finger, and the coordination of the fingers.

Some adaptations of the test protocol and analysis were made in this experiment. Specifically, the keyboard used did not possess the capacity to record velocity. In MIDI, velocity measures the speed at which a key is struck, serving as a rough estimation of the striking force. Furthermore, to facilitate the use of the test in the participants’ homes, we refrained from blindfolding the participants; other than muting the piano sound of the keyboard, no additional auditory masking was used to conceal any additional sound caused by the finger strikes.

We captured and analyzed four metrics: the mean and coefficient of variation (CV) of strike duration, as well as the mean and CV of the interval between strikes. Collectively, these measures function as indicators of the speed and consistency of finger movements.

## 3. Results

The following table provides demographic details about the three recruited participants.

### 3.1. Piano Performance

All participants improved in their piano playing ability throughout the six weeks of training, as judged by improved performance in the selected repertory.

### 3.2. MDS-UPDRS

The changes in six relevant motor subsections of the MDS-UPDRS are presented in Table 2 for the three participants, both before and after the training protocol.

### 3.3. Daily Living Test

The PDQ-39 test was used to compare the quality of life before and after the intervention. In terms of ADL section, the first participant showed an improvement in the dimension score from 20.8 to 12.5, while the second and third participants had the same dimension score before and after the intervention (8.3). In terms of the overall score, the first participant showed almost no change (from 8.3 to 8.2), while the second and third participants showed higher values, corresponding to lower quality of life (participant 2: from 20.8 to 30.5; participant 3: from 3.2 to 7.9).

### 3.4. Behavioral Tests

The results of the nine-hole peg test and the box and block test are shown in Figure 1.

### 3.5. Nine-Hole Peg Test

The results show a significant improvement in overall performance in the NHPT for participants 1 and 3 (see Figure 1a). For participant 3, the performance of the right hand in the NHPT significantly improved, while the performance of the left hand slightly deteriorated. Participant 2’s performance deteriorated. This also corresponds to the significant increase in MDS-UPDRS ratings for participant 2 (from an overall rating of 5 to an overall rating of 10 in the movement-related items measured in this experiment).

As we used an instrumented version of the NHPT, we were able to decompose the total time taken into its components. First, we divided the total time into the time taken to place the pegs and the time to return the pegs to the receptacle. During placing, there is the container (time to take the peg from the container), transport (moving the hand from the receptacle to starting to insert the peg), insert (inserting the peg in the hole), and return (returning the hand to the receptacle). Removing is deconstructed into transport (taking the hand from the receptacle to the peg), removal (removing the peg from the hole), and return (returning the peg to the receptacle).

While Figure 2 suggests changes pre- and post-training in all components, it seems the various placing stages (return, insert, and transport) are dominant contributors to the overall change, especially in participants 1 and 3. In particular, the placing:insert stage that emphasizes dexterity (marked as yellow in the chart) seems to have decreased in participants 1 and 3 but not in 2. A closer look at each participant separately reveals the following details: In participant 1, improvement was noted in both hands. The main contributors to the overall improvement in both cases are the placing stages (return, insert, and transport). In participant 3, the right-hand performance improved significantly, while the left-hand performance slightly deteriorated (a deterioration also reflected in the MDS-UPDRS scores). However, it is interesting to note that the dexterous stages, such as the placing:insert and placing:container stages, improved in both hands and that the overall slower performance of the left hand is a result of an increase in the non-dexterous stages of the NHPT, possibly reflecting slower overall movement of the left hand.

For participant 2, the increase in overall performance in the left hand seems to result mainly from a change in the placing:return stage, while the right-hand performance seems to be affected mainly by placing in the container.

### 3.6. Box and Block Test

The results show stable performance in the BBT for participants 1 and 3 (see Table 3). Participant 2 had significantly improved results in both hands. Considering the progressive nature of PD, maintaining the same result over time is also a positive outcome.

### 3.7. Uncontrolled Manifold Analysis (Pressing Task)

The results of the finger pressing task (see Figure 3) suggest an overall improvement for all participants, which is also reflected by the considerable decrease in straight-line deviation (i.e., accuracy) for all participants. A closer look at the breakdown of Δ*v* into *v_good_* and *v_bad_* reveals a decrease in bad variance for all participants, as well as an increase in good variance for participants 2 and 3, and a slight decrease in good variance for participant 1.

### 3.8. Quantitative Digitography (QDG)

Due to technical problems with the recording, data from one of the participants (participant 2) were not available for analysis. For the QDG, we extracted the duration of the strike and the interval between strikes and calculated the mean and coefficient of variation of these two quantities. We did not calculate the velocity (volume) of the keypress, as in the original publication of the QDG [30], because the MIDI piano we used does not record velocity. The results are summarized in Figure 4. Participant 1 showed improvement in both hands in terms of the mean strike duration and the interval between strikes but showed an increase in CV for both of these quantities (except for the left hand in the CV of the interval between strikes, which reduced slightly). In contrast, participant 3 showed a small decrease in mean strike duration and mean interval between strikes (apart from right-hand strike duration) but a reduction in the CV for both hands and measures.

## 4. Discussion

In this pilot study, we described a piano-based training protocol for people with Parkinson’s disease (PwP) and demonstrated how different sensors can be used to evaluate changes in dexterity in this population that result from this intervention. While tests that are commonly used in clinical practice, such as the MDS-UPDRS or the box and block test, can quantify some changes in dexterity, small changes may not lead to changes in the measure (e.g., when using the MDS-UPDRS), or these measures may be susceptible to compensation [38]. In contrast, sensor-based methods such as the instrumented nine-hole peg test, the uncontrolled manifold analyses, and Quantitative Digitography, which were used in this study, are able to quantify small changes in performance, differentiate between compensation and recovery [39], and potentially provide an interpretation as to which aspects of movement are leading to changes in dexterity (for better or worse). This information can be useful for analyzing the outcomes of clinical trials and assisting clinicians to appropriately select treatments.

The integration of sensor technology also has the potential to provide real-time feedback on performance, allowing for individualized customization and adjustment of training protocols. This is important for use with PwP, given that the symptoms vary considerably across different people and the rate of progression of the disease also varies widely [40]. One of the measures used here, Quantitative Digitography [30], is run using the same piano keyboard as is used for the training, making it a particularly attractive choice for tracking changes in dexterity over time.

To quantify the feasibility of this approach, we tested three people with Parkinson’s disease (PwP), who performed a six-week, piano-based training program. As well as improving in their playing of the piano, they also showed some improvements in dexterity, in terms of their scores on aspects of the nine-hole peg test and the synergy index in a multi-finger force pressing task. In terms of quality of life measured by the ADL section of PDQ-39 test, one participant showed an improved score and the other two participants had the same score, while their overall scores showed an increase (corresponding to a lower quality of life).

For one participant, the PDQ-39 assessment indicated a stable overall rating with a notable two-point improvement in the ADL section. This enhancement surpasses the minimally important improvement documented in prior research [41]. Conversely, the other two participants exhibited an uptick in their PDQ-39 overall scores, suggesting a decline in their perceived quality of life. Interestingly, despite the rise in their overall scores, the ADL section within the PDQ-39 for these participants showed no change. Given the progressive trajectory of PD, it is typically expected for scores to increase over time [42]. However, it is noteworthy that this increment in the overall score was not attributed to the ADL score, which predominantly involves dexterity capabilities. Piano training in PwP may also be beneficial for non-motor-related cognitive performance. For example, a recent study showed improved performance on the Stroop test as a result of 30 h of training on the piano for PwP [43].

In the box and block test (BBT), one participant exhibited an improved score, whereas the other two participants maintained their previous scores. Again, given PD’s progressive nature, a gradual decrease in BBT scores is generally anticipated over time [31]. Therefore, the preservation of consistent scores over a period of time is noteworthy. Notwithstanding this evidence, it is worth emphasizing that the BBT primarily evaluates gross manual motor skills. In contrast, the experimental training in this study predominantly targeted fine manual motor skills. Consequently, the BBT might only capture skills tangentially related to the training provided in this study. Notably, in research on constraint-induced movement therapy [44] and a VR variant of the test [45], an increase in BBT scores was observed among PD patients. However, those interventions involved gross and fine motor skills, distinguishing them from this study’s fine motor skill focus.

In this study, we used an instrumented version of the nine-hole peg test (NHPT), which allowed us to deconstruct the total time taken into seven components—four while placing the pegs, and three while removing the pegs. Some of these components rely on more gross motor skills in moving the hand to the right place (placing: transport, placing: return, removing: transport, and removing: return), while others are more reliant on fine motor skills—placing and removing the pegs from the holes or the container (placing: container, placing: insert, and removing: removal). Among the various tests we employed, the NHPT distinctly emphasizes fine motor skills, particularly evident in the parts requiring peg manipulation—such as removing and inserting the pegs. Further enriching our analysis, the more granular insight into various movement components that the instrumented version of the NHPT affords allowed us to estimate the relative contribution of the dexterous and non-dexterous motions to the overall performance. Intriguingly, for participants 1 and 3, who exhibited enhanced overall NHPT performance, the improvements were predominantly in the dexterity-related segments of the NHPT (marked as placing: container, placing: insert, and removing: removal in the graph) rather than in the gross-movement segments. Furthermore, despite participant 2 demonstrating a reduced overall performance in the NHPT, the increase in task completion time was mainly attributed to the transport and placement into the container stages and not the dexterous phases. Such a manifestation is likely the result of increased bradykinesia. Thus, the breakdown into various movement components suggests that in all participants the dexterous segments improved or were less negatively affected compared to the non-dexterous segments.

In terms of the uncontrolled manifold (UCM) analysis, an increase in the synergy index was observed for all three participants. An increase in the synergy index corresponds to an increase in stability [46]. The synergy index is known to be lower in PwP [47,48] and did not show a change after a dose of levodopa in levodopa-naïve PwP [48], although they showed that a single dose did cause an increase in both good and bad variance. The improvement observed in this study of the synergy index suggests that stability improved as a result of the training. It demonstrates that there is room for improvement in stability as a result of appropriate training, e.g., focusing on finger differentiation and coordination.

As an additional measure of performance, we used Quantitative Digitography (QDG) [30]. While QDG uses the piano as the measurement device, it does not evaluate how well participants learned to play the piano in general but rather focuses on particular finger gestures, which are somewhat similar to the finger tapping task in the MDS-UPDRS but allow a finer resolution of measurement compared to the relatively gross scale of the MDS-UPDRS. In the QDG, a reduction in the strike duration or interval suggests improved overall performance, whereas a reduction in the coefficient of variance (CV) corresponds to a more regular movement (or less freezing). Unfortunately, we only managed to record the QDG from two participants due to technical problems. In these two participants, we observed different outcomes—participant 1 showed decreases in the mean values but increases in variability, whereas participant 3 showed mixed values for the mean values but decreases in the variability. These differences between the start and end of training may have been due to the higher baseline values for variability for participant 3, allowing more room to improve. A benefit of the QDG in evaluating changes in dexterity due to piano training is that it does not require additional hardware and can, in theory, be measured at home.

In summary, we employed a comprehensive suite of tests designed to assess various dimensions of manual functionality, encompassing stability (UCM), fine movement (NHPT), gross movement (BBT), and regularity (QDG), and combined sensor-based and manual tests. The most significant benefits of the training were observed in the UCM and NHPT tests, indicating marked improvements in manual stability (a measure influenced by finger differentiation and coordination) and enhancements in dexterous aspects of finger manipulation post-training.

Our findings underscore the importance of employing a differentiated test battery to effectively capture the nuanced benefits of dexterity training. Some tests, such as the BBT, were less affected by the training, possibly due to their focus on less relevant aspects of manual movement (gross vs. fine). Furthermore, the improvements in the dexterous segments of the instrumented NHPT would likely have been obscured by decrements in gross manual segments for certain participants in the non-instrumented version of the test.

While additional research is warranted, the current study suggests that piano training may indeed enhance some aspects of dexterity performance in PwP, and that such improvements could, in turn, contribute to the maintenance and improvement of ADL abilities, as suggested by the ADL scores in the PDQ-39.

### 4.1. Future Directions

While this study offers encouraging insights into the potential advantages of piano training, additional research involving a more diverse participant pool and a wider array of PD manifestations and severity is essential to validate and further establish these findings. A randomized controlled trial could provide further clarity regarding the robustness and replicability of our outcomes. Moreover, in comparison to other forms of physical training in PwP, such as dance [49] or Tai Chi [50], the duration of our training regimen was relatively brief. Extended training periods could reveal amplified or distinct benefits that were not evident in our preliminary study.

### 4.2. Limitations

This study serves as a preliminary, open-label pilot study, which demonstrated techniques for training and evaluation. Although the findings are promising, they warrant further investigation. Specifically, due to this study’s small participant pool it should be viewed as an indicative rather than a conclusive sign of potential benefits. Expanding the participant pool in terms of both their number and diversity in Parkinson’s disease (PD) stages and types within the setting of a randomized controlled experiment with an age-matched control group of non-PD individuals could provide clearer results as well as a more nuanced understanding of piano training’s benefits and shortcomings and how this is affected by both ageing and PD.

Among the tests conducted were cognitive assessments that were only administered at the start as an inclusion criterion. Incorporating a post-intervention cognitive test in a future large-scale study protocol would be advantageous, considering that musical interventions have been demonstrated to positively influence cognitive functions in PwP [43].

The study curriculum combined standardized elements with individual customization to accommodate participants’ musical preferences and needs. While this methodology offers its own merits, employing a more uniform curriculum would facilitate tighter control over this variable, enhancing the study’s internal validity.

The choice of MIDI keyboard for this study was influenced by its portability in terms of weight and size. However, it lacked velocity sensitivity, the MIDI equivalent of touch-based loudness control. Using a touch-sensitive keyboard could potentially offer the participants a more enriched learning experience. Correspondingly, the QDG test in our experiment did not incorporate velocity measurements, which would have added another dimension to the assessment of participants’ performance.

Finally, the study’s six-week duration is relatively brief compared to other physical training interventions for rehabilitation. An extended training period may reveal additional or more pronounced benefits that were not discernible within the confines of this initial study.

## Figures and Tables

**Figure 1 sensors-24-03318-f001:**
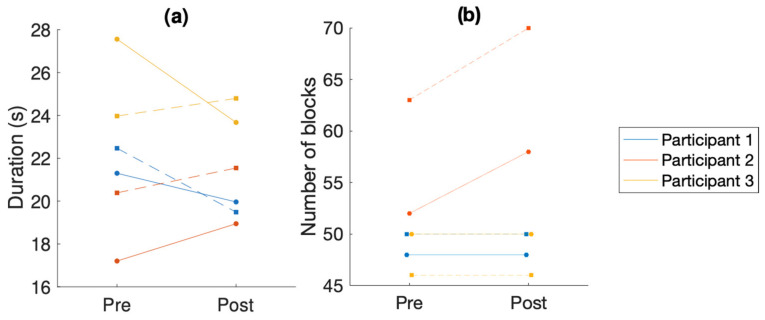
Comparison of overall performance in the behavioral tests. (**a**) Time taken to complete the nine-hole peg test (NHPT). (**b**) Number of blocks moved over the barrier in the box and block test. In both graphs, the solid line is the left hand and the dashed line is the right hand. Each color is a different participant.

**Figure 2 sensors-24-03318-f002:**
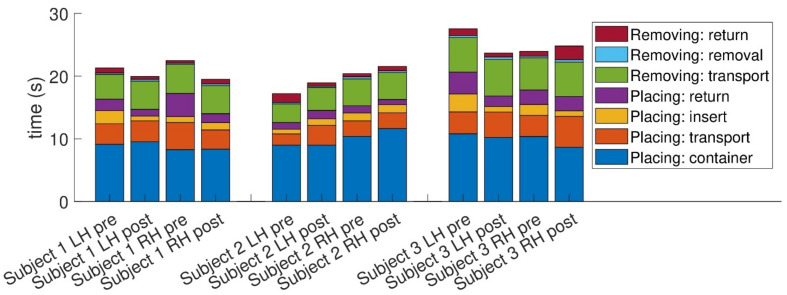
Decomposition of the time taken to perform the nine-hole peg test into components. The definition of the different components is provided in the text.

**Figure 3 sensors-24-03318-f003:**
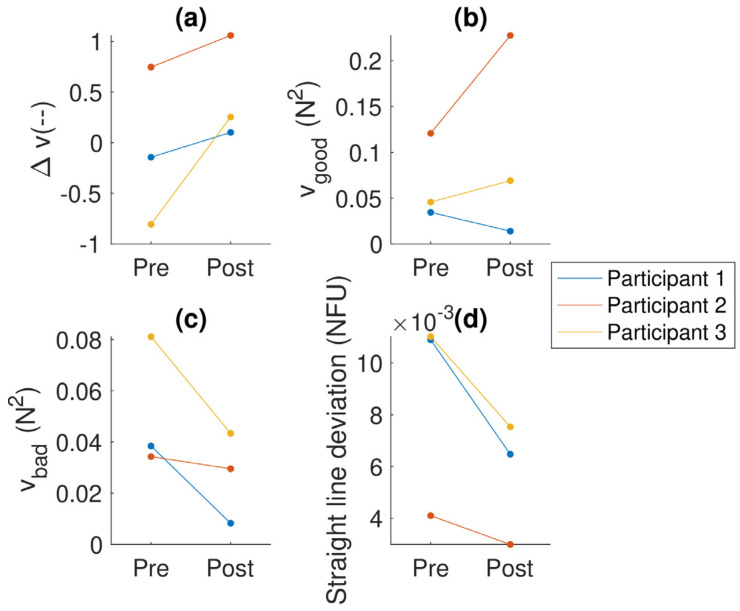
Analyses of the finger pressing task: (**a**) the synergy index (Δ*v*); (**b**) the amount of good variance (*v_good_*), which does not affect the outcome variable; (**c**) the amount of bad variance (*v_bad_*), which does affect the outcome variable; and (**d**) straight line deviation—a measure of how well they controlled the total force (as shown on the screen). The colors indicate the participant (the same colors are used for each participant across the figures).

**Figure 4 sensors-24-03318-f004:**
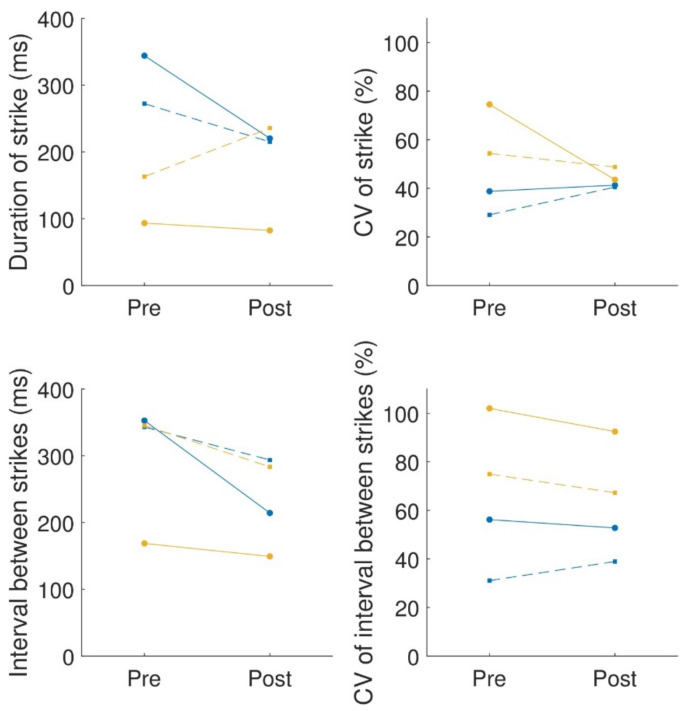
QDG summary: mean (left column) and coefficient of variance (CV) (right column) of the strike duration (first row) and interval between strikes (second row). The solid lines connecting the circles are for the left hand; the dashed lines connecting squares are for the right hand. Each color is a different participant.

**Table 1 sensors-24-03318-t001:** Participant demographic details.

Participant	Age	Sex	Dominant Hand	Mini Mental Score	Modified Hoehn and Yahr Scale	Disease Duration (Years)	Duration between Pre- and Post-Tests
1	41	F	R	30	2	4	10.2 weeks
2	52	M	R	29	2.5	4	16.2 weeks
3	71	M	L	29	2	10	16.2 weeks

**Table 2 sensors-24-03318-t002:** Scores for six relevant motor subsections of the MDS-UPDRS. The scores are given for both the pre-test (Pre) and post-test (Post) and are for the dominant hand. Higher scores indicate more impairment.

Participant	Pre/Post	Part II				Part III		Overall
		2.4	2.5	2.6	2.7	3.3	3.4	
1	Pre	0	1	0	2	1	0	4
	Post	0	0	0	2	1	1	4
2	Pre	0	1	0	0	2	2	5
	Post	1	1	1	1	3	3	10
3	Pre	0	0	0	0	0	1	1
	Post	1	1	0	0	1	1	4

**Table 3 sensors-24-03318-t003:** Scores on the box and block text before (Pre-) and after (Post-) training, for the left (LH) and right (RH) hands. The score is the number of blocks successfully moved over the barrier in one minute.

Participant	Pre-LH	Pre-RH	Post-LH	Post-RH
1	48	50	48	50
2	52	58	63	70
3	50	46	50	46

## Data Availability

The data and software used are available on Figshare: https://doi.org/10.6084/m9.figshare.23994768 (accessed on 2 April 2024).

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
