# Peer review of "Quantifying Changes in Dexterity as a Result of Piano Training in People with Parkinson’s Disease"

_sensors, 2024, doi:10.3390/s24113318_

Round 1

Reviewer 1 Report

Comments and Suggestions for Authors

The manuscript tested three participants on a six-week custom piano-based training program to evaluate the dexeterity of people with Parkinson's disease. A diversity of methods are employed for quantitative evaluation using various sensors. However, the conclusion drawn from this study is ambiguous. If it is to demonstrate the efficacy of the piano-based training program in improving the dexeterity, the number of participants is far too small to give reliable insights. On the other hand, if the study focuses on the design of sensor-based measurement of dexeterity, the advantage of these proposed methods should be discussed in more details and prove why they are preferrable to or consistent with the traditional measurement. Unfortunately, the manuscript fails to justify its values in either of these two aspects. 

Author Response

We thank the reviewer for their feedback on the article. We agree that the sample size and lack of a control group means that we cannot make definitive claims about the efficacy of the protocol. Rather, we present the study as a pilot, and have emphasized the advantages of using sensor-based technologies for tracking changes in dexterity in such a study, and have made available all the tools we used. We have added two paragraphs to the start of the discussion focusing on this (lines 347-366):

“In this pilot study, we described a piano-based training protocol for people with Parkinson’s disease (PwP) and demonstrated how different sensors can be used to evaluate changes in dexterity in this population that result from this intervention. While tests that are commonly used in clinical practice, such as the MDS-UPDRS or the box and block test, can quantify some changes in dexterity, small changes may not lead to changes in the measure (e.g., when using the MDS-UPDRS), or these measures may be susceptible to compensation [38]. In contrast, sensor-based methods such as the instrumented nine hole peg test, the Uncontrolled Manifold analyses, and Quantitative Digitography, which were used in this study, are able to quantify small changes in performance, differentiate between compensation and recovery [39], and potentially provide an interpretation as to which aspects of movement are leading to changes in dexterity (for better or worse). This information can be useful for analyzing the outcomes of clinical trials, as well as for clinicians for appropriately selecting treatments.

The integration of sensor technology also has the potential to provide real-time feedback on performance, allowing for individualized customization and adjustment of training protocols. This is important for use with PwP, when the symptoms vary considerably across different people, and the rate of progression of the disease also varies widely [40]. One of the measures used here, Quantitative Digitography [30], is run using the same piano keyboard as is used for the training, making it a particularly attractive choice for tracking changes in dexterity over time."

Reviewer 2 Report

Comments and Suggestions for Authors

The paper is well written and methodologically well structured. However, there are some aspects that could be improved. The introduction lacks a description of innovative interventions currently used for rehabilitation in patients with Parkinson's disease, which would provide a current overview of the situation. For instance, I suggest mentioning two protocols describing innovative interventions for Parkinson's patients: The latter also produced initial results [Maranesi et al. (2022). The Effect of Non-Immersive Virtual Reality Exergames versus Traditional Physiotherapy in Parkinson’s Disease Older Patients: Preliminary Results from a Randomized-Controlled Trial. International journal of environmental research and public health, 19(22), 14818].

In my opinion, the description of the sample and its corresponding table should be relocated to the Results section. Additionally, there is a missing reference to the Short version of the MDS-UPDRS scale in line 134 and following. Lastly, the limitations section should include a mention of the limited sample size considered in the study.

Author Response

We thank the reviewer for their feedback and suggestions for improvement of the manuscript.

Reviewer comment: The introduction lacks a description of innovative interventions currently used for rehabilitation in patients with Parkinson's disease, which would provide a current overview of the situation. For instance, I suggest mentioning two protocols describing innovative interventions for Parkinson's patients: The latter also produced initial results [Maranesi et al. (2022). The Effect of Non-Immersive Virtual Reality Exergames versus Traditional Physiotherapy in Parkinson’s Disease Older Patients: Preliminary Results from a Randomized-Controlled Trial. International journal of environmental research and public health, 19(22), 14818].

Response: We expanded this section of the paper based on this comment (lines 53-60). We note that while the comment mentioned two protocols, only one citation was given.

"When hypothesizing the beneficial factors that render music an effective rehabilitative tool, two elements stand out. First, the rich multisensory feedback provided by playing music may serve as an enhanced biofeedback mechanism, facilitating rapid error detection and correction. Second, the engaging nature of music playing likely contributes to the success of music-based interventions. Parallel to these, a line of innovative interventions incorporate other techniques, such as the use of VR-based exergames to improve motor capabilities in Parkinson’s disease patients, which have been shown to enhance balance and gait [11] and improve dexterity [12]."

Reviewer comment: There is a missing reference to the Short version of the MDS-UPDRS scale in line 134 and following.

Response: We selected relevant parts of the MDS-UPDRS scale to focus on relevant changes in dexterity due to the protocol, and this indeed is not an official “short” version.  We corrected the text by changing “short MDS-UPDRS” in lines 244, 247, and the abstract (line 19) to “selected relevant items from the motor parts of the MDS-UPDRS”.

Reviewer comment: The limitations section should include a mention of the limited sample size considered in the study.

Response: We added the following to the limitations section (line 476) : “Specifically, the study's small participant pool should be viewed as an indicative rather than a conclusive sign of potential benefits.”

Reviewer comment: The description of the sample and its corresponding table should be relocated to the Results section

The participant demographic table has been moved to the Results section (line 238).

Reviewer 3 Report

Comments and Suggestions for Authors

Would recommend in discussion placing limitations of cognitive testing was not performed after testing and recommend performing in a larger study.

Second limitation should mention very small sample size but larger samples should be more informative.

stidy design was good and in a larger study would compare to a control of non pd individuals age matched. You cna put that in the paper 

good job. I recommend publishing with these minor changes suggested.

Author Response

We thank the reviewer for their constructive feedback on the article.

Reviewer comment: Would recommend in discussion placing limitations of cognitive testing was not performed after testing and recommend performing in a larger study.

Response: Thank you for this comment. The following has been added to the limitations section (lines 483-486):

"Among the tests conducted were cognitive assessments that were only administered at the start as an inclusion criterion. Incorporating a post-intervention cognitive test in a future large-scale study protocol would be advantageous, considering that musical interventions have been demonstrated to positively influence cognitive functions in PwP [38]."

Reviewer comment: Second limitation should mention very small sample size but larger samples should be more informative.

Response: We added the following sentence (line 476):

"Specifically, the study's small participant pool should be viewed as an indicative rather than a conclusive sign of potential benefits."

Reviewer comment: study design was good and in a larger study would compare to a control of non pd individuals age matched. You cna put that in the paper 

Response: we added this sentence to the limitations (line 477):

"Expanding the participant pool in terms of both their number and diversity in Parkinson's Disease (PD) stages and types within the setting of a randomized controlled experiment with an age-matched control group of non-PD individuals could provide clearer results as well as a more nuanced understanding of piano training's benefits and shortcomings and how this is affected by both ageing and PD."

Round 2

Reviewer 1 Report

Comments and Suggestions for Authors

The authors provide justifications as a reply to my comments in the first-round review, however, this does not address my concerns listed in the first-round review report. The study itself may be a reasonable pilot research report but is still below the bar of acceptance to Sensors. There is neither technical/algorithmic contribution nor reliable insight into the application scenarios.